# Reliable QoE Prediction in IMVCAs Using an LMM-Based Agent

**DOI:** 10.3390/s25144450

**Published:** 2025-07-17

**Authors:** Michael Sidorov, Tamir Berger, Jonathan Sterenson, Raz Birman, Ofer Hadar

**Affiliations:** School of Electrical and Computer Engineering, Ben Gurion University of the Negev, Be’er Sheba 8499000, Israel; tamirber@post.bgu.ac.il (T.B.); sterenjo@post.bgu.ac.il (J.S.); birmanr@post.bgu.ac.il (R.B.); hadar@bgu.ac.il (O.H.)

**Keywords:** Large Multimodal Models, encrypted traffic, quality of experience, machine learning, video conferencing

## Abstract

Face-to-face interaction is one of the most natural forms of human communication. Unsurprisingly, Video Conferencing (VC) Applications have experienced a significant rise in demand over the past decade. With the widespread availability of cellular devices equipped with high-resolution cameras, Instant Messaging Video Call Applications (IMVCAs) now constitute a substantial portion of VC communications. Given the multitude of IMVCA options, maintaining a high Quality of Experience (QoE) is critical. While content providers can measure QoE directly through end-to-end connections, Internet Service Providers (ISPs) must infer QoE indirectly from network traffic—a non-trivial task, especially when most traffic is encrypted. In this paper, we analyze a large dataset collected from WhatsApp IMVCA, comprising over 25,000 s of VC sessions. We apply four Machine Learning (ML) algorithms and a Large Multimodal Model (LMM)-based agent, achieving mean errors of 4.61%, 5.36%, and 13.24% for three popular QoE metrics: BRISQUE, PIQUE, and FPS, respectively.

## 1. Introduction

Instant messaging-based video call applications (IMVCAs) and Video Conferencing Applications (VCAs) have become essential tools in modern communication, transforming how we connect over distances. The COVID-19 pandemic has further accelerated the adoption of video conferencing across various domains, including work, healthcare, education, and personal interactions. IMVCAs have facilitated face-to-face video communication, primarily serving interpersonal communication needs. In addition, Their diverse features have enabled them to penetrate markets traditionally dominated by VCAs. For instance, a study highlighted WhatsApp’s utility in education during the COVID-19 pandemic, demonstrating its effectiveness by leveraging various features, including video calls, group chats, document sharing, and more [1,2,3,4,5,6]. As real-time video communication continues to gain popularity, ensuring a high Quality of Experience (QoE) for users across the variety of VCAs and IMVCAs is crucial. Studies [7,8] have shown that network conditions, such as throughput, latency, and packet loss, directly impact user-perceived experience. Traffic patterns can vary significantly, leading to various disruptions across different applications. Therefore, it is critical for Internet Service Providers (ISPs) to diagnose, predict, and address, hopefully ahead of time, the different issues related to the QoE of their users as part of efficient network functionality.

Due to the rise in the demand for privacy over the Internet, most of the traffic that is being transmitted over the Internet is encrypted. In addition, many IM applications, including WhatsApp, use an end-to-end encryption. While content providers (e.g., WhatsApp) have no difficulty assessing the QoE of their users (as they encrypt the data), ISPs must estimate user QoE using passive measurements of the encrypted traffic traversing their network.

In this study, we address the challenges faced by ISPs in evaluating user QoE using encrypted traffic data. We propose an approach for evaluating video quality metrics using an LLM-based agent that is able to predict the QoE of users based only on features that may be easily extracted from Internet traffic.

We validated our approach on WhatsApp IMVCA, and we compare it against models that are frequently used in this domain and provide SOTA results for QoE evaluation. WhatsApp neither provides quality metrics nor publicly discloses its communication protocol, emphasizing the necessity of our approach.
Our contributions are as follows:
We develop an LLM-based framework for estimating video QoE metrics from encrypted traffic.We conduct a pioneering study on WhatsApp, creating datasets of encrypted traffic and corresponding QoE metrics, thereby closing the gap in IMVCAs.We provide a thorough analysis of the features that may be easily extracted from encrypted Internet traffic, serving as an assessment of the QoE of users who participate in the conversation.Our final contribution is releasing the dataset together with the code for feature extraction and prediction in order to stimulate future research in this domain.

The remainder of this paper is structured as follows: Section 2 reviews related work in the field of video QoE estimation. Section 3 details our proposed methodology. Section 4 presents the experimental setup and evaluation results. Section 5 discusses the implications of our findings, and Section 6 concludes this paper, with future research directions provided.

## 2. Related Work

Recent years have seen an expansion in research on video QoE inference from encrypted traffic. For example, Dubin et al. [9] demonstrated an exceptional accuracy of 97% in the prediction of QoE in YouTube streaming services by analyzing the timing of line-traversing packets. In a latter study, Gutterman et al. [10] trained a Decision Tree (DT) algorithm in an attempt to extended this work by trying to predict two additional metrics, viz., buffer warning and video state, which, by their assumption, should be indicative of the overall QoE of users. In this course, Mazhar and Shafiq in [11] trained a DT classifier to predict the QoE of the users on YouTube, but their method had low granularity (i.e., “good”/“bad” QoE) and cannot be used in an online setting. Wasserman et al. [12] predicted standard QoE Key Performance Indicators (KPIs), viz., stalling, initial delay, video resolution, and average video bitrate, using encrypted YouTube traffic. For this purpose, they used the recently developed Large Language Models (LLMs), which were initially developed for textual data analysis.

This approach was latter extended to work with data from other domains by introducing the Large Multimodal Models (LMMs) that are able to be efficiently applied on data from other domains, such as images [13] and time series [14].

LMMs were recently used by Ginige et al. [15] to expand the prediction of hazardous Internet traffic from a closed setting (i.e., where the number of classes is determined in advance) to a more realistic open setting, where the types of the possible hazards are not known ahead of time. They operated on encrypted data and showed that, by the correct employment of LMMs, substantial improvements over other models used for this type of problem may be achieved.

Another study that employed an LMM-based agent is a recent work performed by Shaham et al. [16], where the authors showed impressive performance in the task of feature interpretability and failure analysis. In their experiments, the trained, in an unsupervised way, an LMM-based agent to be able to choose correct features for experiments and then automatically fit models on these features. This work shows the great potential of LMMs and the special use cases they may be used for.

## 3. Methodology

In this section, we describe our methodology used to predict the QoE metrics of the IMVCA sessions. For this purpose, we chose four models that are widely used in previous studies for regression problems in tabular data, viz., Random Forest (RF), Support Vector Machine (SVM) [17], Extreme Gradient Boosting (XGBoost) [18], and Categorical Boost (CatBoost) [19]. In addition, we designed a pipeline to employ the current LLM-based agent to perform the same prediction task. Next, we will describe each of the algorithms used in this research and its main features.

### 3.1. Random Forest

Random Forest (RF) combines multiple Decision Tree (DT) models into a single system that can be used for both regression and classification tasks. It operates in three main steps: (1) Bootstrapping: The original dataset is sampled with replacements to create multiple different subsets. (2) Model Training: A separate DT is trained on each dataset subset created in the previous step, resulting in a collection of “weak learner” models that vary slightly due to differences in training data. (3) Aggregating: For classification tasks, the final prediction is determined by a majority vote across all DTs, while for regression tasks, the predictions are calculated as an average of all predictions of the “weak learners”.

### 3.2. Support Vector Machines

Support Vector Machines (SVMs) are supervised machine learning algorithms that are primarily used for classification tasks, though they can also be applied to regression. In their basic form, SVMs construct a linear model, assuming that data points are linearly separable. However, this approach can be extended to handle non-linear relationships by employing kernel functions (a technique known as the “kernel trick”), which projects the data into a higher-dimensional space where linear separation is possible. At its core, an SVM fits a line (in 2D) or a hyperplane (in higher dimensions) that best separates the classes by maximizing the margin between them. SVMs are especially effective with high-dimensional data and are relatively robust to overfitting. However, they are sensitive to hyperparameter tuning and can be difficult to interpret in dimensions beyond three.

### 3.3. Extreme Gradient Boosting

Extreme Gradient Boosting (XGBoost) is an ensemble technique that is similar to RF but with a key difference: While RF trains DTs in parallel on different subsets of data, XGBoost trains the “weak learners” (typically a DT) sequentially, teaching them to correct the errors made by the previous ones. It employs techniques such as 
$L_{1}$
 (lasso) and 
$L_{2}$
 (ridge) regularization to reduce overfitting and uses tree pruning, parallelization, cache optimization, and efficient data structures for increased performance. In addition, it handles missing data automatically, and it uses custom loss functions for a flexible model design.

### 3.4. Categorical Boosting (CatBoost)

Categorical Boosting is a strong algorithm developed by Yandex for handling tabular data. This algorithm falls under the class of Gradient Boosting, and it is built on Decision Tree (DT) ensembles, like XGBoost, but it performs better over labels with several classes due to modifications made to make the DT more symmetric. Although this algorithm usually takes longer to train, it demonstrates better results over tabular data. Unlike XGBoost, CatBoost does not require hyperparameter tuning. It also handles regression problems very well, as any regression problem may be generalized to a classification with multiple classes.

### 3.5. Large Language Models (LLMs)

LLM is a special type of transformer model that is trained on a very large dataset of text that enables it, in a very efficient way, to answer text-based queries. The main strength of the LLM is the vast number of parameters it employs—while GPT-1 used 117 M parameters, the newest iteration of GPT, GPT-4, uses ∼1 T parameters. The basic principle that is used in the transformer ANN is the mechanism of attention, which concentrates the model on relevant information and discards irrelevant information. In other words, self-attention captures long-term dependencies rather than just local ones. Self-attention is computed using the tokens of the input prompt, and it is calculated using the following equation:
(1)
$$Attention\left(Q,K,V\right)=softmax\left(\frac{Q\cdot K^{T}}{\sqrt{d_{k}}}\right)\cdot V$$

where *Q* is called the “query”, *K* denotes the “keys”, and *V* denotes the corresponding “values”. The 
$Q\cdot K^{T}$
 measures the similarity between tokens, which are squashed into weights by the softmax function. Then, a weighted sum of the values is produced via multiplication by the *V* term.

The normalization term 
$\sqrt{d_{k}}$
 is used to stabilize the gradient froq and the learning procedure in cases of high-dimensional input, and it prevents the softmax function from saturating. As the transformer [20] architecture employs layer normalization initialized with Xavier weight initialization (with weights initialized as 
$W_{i,j}∼N\left(0,\frac{2}{n_{in}+n_{out}}\right)$
 [21]), it is valid to assume that the inputs have a zero mean and a standard deviation of 1, so the dot product between some query 
$q_{i}$
 and some key 
$k_{j}$
 will be
(2)
$$\alpha =\sum_{n=1}^{d_{k}}q_{jn}k_{jn}^{T}$$

and the variance of 
α
 will be
(3)
$$Var\left(\alpha \right)=Var\left(\sum_{n=1}^{d_{k}}q_{jn}k_{jn}^{T}\right)=\sum_{n=1}^{d_{k}}Var\left(q_{jn}k_{jn}^{T}\right)$$


Since we suppose that 
$\forall n\in \left[1,d_{k}\right],Var\left(q_{in}\right)=Var\left(k_{jn}\right)=1$
, we get
(4)
$$Var\left(\alpha \right)=\sum_{n=1}^{d_{k}}1=d_{k}$$

which gives us a standard deviation of 
$\sqrt{d_{k}}$
. This situation may saturate the 
Softmax(·)
 function and make it put all weights on one particular entry and 0 for all other entries. This makes the gradients vanish and prevents the model from learning. To prevent this from happening, the dot product is normalized by 
$\sqrt{d_{k}}$
.

### 3.6. Large Multimodal Models (LMMs)

LLMs are breakthrough methods that are not less significant than the introduction of Convolutional Neural Networks (CNNs) in 2012 [22]. The main difference in LMMs compared to the LLM is the ability of the former to be efficiently applied on non-textual data, like images, time series, audio, etc. Some of the best LMM models as of today are ChatGPT-4o (Omni) by OpenAI, Gemini 1.5 by Google DeepMind, and Flamingo by DeepMind, which can handle the input of text, images, or audio data; or BLIP-2 by Salesforce, Fuyu by Adept, and Kosmo-2 by Microsoft, which are limited to text and image data.

The LMMs function by using a combination of an LLM backbone, which interprets the queries of the user and encodes them; a data-specific encoder, which is used for non-textual input transformation; and a fusion mechanism used to blend all inputs together, transforming them into a form that can be used for output generation.

## 4. Data

In this section, we discuss the dataset used for the experiments, including its collection, exploration, and feature engineering.

### 4.1. Data Collection

The data was collected in a controlled manner by instantiating calls via WhatsApp IMVCAs, lasting 4 min each. In each session, two roles were designated to the participants, viz., transmitter and receiver, where the former used a cellular device while the latter was connected through a WhatsApp’s desktop application that ran on a PC in the lab. The features were extracted using the WireShark application (see Figure 1), and the extracted features are listed in Table 1.

In addition, to be able to quantify the QoE of the session on the *receiver* side, we extracted the two types of screen captures as follows: (1) a high-resolution screenshot of the image seen by the *receiver* in a lossless format with a frame rate of 1 FPS and (2) a small portion of the screen with a low resolution but high frame rate. The first screen capture was used to evaluate the visual QoE by the means of two no-reference QoE metrics, viz., BRISQE and PIQUE. The second type of capture enabled us to extract the third label of interest, which measures the temporal resolution of the interaction, i.e., the FPS.

As the BRISQUE, and PIQUE QoE metrics use the spacial resolution of the image for their quality assessment, the type of sensor used by each side in the session plays a critical role in the final QoE. In our experiments, we used several models of cellular devices, as well as laptop models and ICams, to make our results more general. The models of the devices, together with the photo sensor’s specifications, are presented in Table 2.

To simulate a realistic setting, we introduced three types of limiting parameters on the *receiver side*, viz., packet loss, bandwidth limit, and sudden bandwidth drops, as described in Table 3. To avoid bias towards a certain type of limitation, the same number of sessions was recorded from each type, and the limitations of packet loss and bandwidth drop were evenly distributed between the different values.

### 4.2. Feature Extraction

Similarly to Sharma et al. [23], we extracted two types of features features, viz., features related to the size of the packets and ones related to the packet inter-arrival times (PIATs), as presented in Table 4.

### 4.3. Video Stream Classification

The classification process applied to the collected data consisted of two steps. First, detect the packets related to the specific application stream. Second, isolate the video stream data from these packets. We identify the video application traffic by the packets’ IP addresses and subsequently extract only the video packet data, disregarding all other packet types. Previous studies on Zoom [24,25] have distinguished media types by utilizing RTP headers, specifically the RTP payload type, to classify packets into different categories, such as videos, screen sharing, and audio. In study [23], Taveesh Sharma et al. differentiated packets by size based on the assumption that audio packets require fewer bits than video packets. Adopting this approach, we classify packets based on their size, ensuring the method’s applicability even when the RTP headers are encrypted or the payload type is modified by the application’s protocol changes. Upon investigating our WhatsApp dataset, we found that it used RTP payload type (PT) 97 for videos, PT 120 for audio, and PT 103 for video retransmission throughout the entire dataset. The ground truth distribution of packet lengths by RTP payload types in our entire dataset is shown in Figure 2. Our analysis revealed that the most effective classification threshold is 275 bytes, achieving a classification accuracy of 99.99%. Table 5 presents the confusion matrix for video packet classification.

### 4.4. Ground Truth Metrics

#### 4.4.1. Frame Rate

The procedure for determining the frame rate involves the following steps:Creating a Unique Visual Identifier for Each Frame on the Transmitter Side: A screen displaying *S* changing signs is placed in the background of the participant. The signs change at rate *V*, which must be higher than the maximum frame rate of the video application. The number of signs *S* must satisfy 
S≥V·T
, where *T* is the duration of the time slot in seconds. This ensures that each sign appears at most once within a time slot, making it a unique identifier for each frame.Frame Rate Measurement on the Receiver Side: On the receiver’s end, successive window captures are taken of the specific area where the changing sign appears. These captures are performed at a rate *V* and are collected along with their corresponding timestamps. Consequently, there are more captures than frames within the time slot, with some duplication, ensuring that each frame is captured at least once.Associating Frames with Time Slots Based on Arrival Times: At the end of each session, the program processes the collected images in chronological order, retaining only the first capture of each frame by removing all duplicates. Thus, each capture in this list represents a unique frame in the session, ensuring that all frames are represented. The captures are then grouped into time slots.Calculating Frame Rate for Each Time Slot: For each time slot, the number of unique captures associated with it is summed. The frame-per-second rate for time slot *i* is calculated as follows:
(5)
$${\mathrm{fps}}_{i}=\frac{1}{T}\sum_{k\in i}1_{\left[{\mathrm{capture}}_{k}\neq {\mathrm{capture}}_{k-1}\right]}$$

where 
1
 is the indicator function for a new frame, *T* is the duration of the time slot in seconds, *i* is the time slot number, and *k* is the index of the capture.

The procedure is described in Algorithm 1.
**Algorithm 1** Frame Rate Calculation
**Input:** captures, endTimeSlots**Output:** frameRates
N←
 captures.lengthuniqueCaptures 
←[ ]
   *(empty list)*frameRates 
←{}
      *(empty dictionary)***for** endTime in endTimeSlots **do**    frameRates[endTime] 
←0
**end for****for** i in 1:N **do**   **if** captures[i].image ≠ captures[i-1].image **then**      uniqueCaptures.append(captures[i])   **end if****end for****for** capture in uniqueCaptures **do**    frameRates[⌈ capture.timeStamp ⌉] += 1**end for****return** frameRates


To validate the proposed method, we generated 30 distinct videos, each with a constant frame rate ranging from 1 to 30 frames per second (FPS). By concurrently playing these videos and executing the frame rate detection program on the same device, we eliminated network effects, ensuring precise frame rate display on the screen. The validation results demonstrated high accuracy across the entire tested FPS range, with a Mean Absolute Error (MAE) of 0.06 FPS and an accuracy of 97.5% within an error tolerance of 5% relative to the ground truth. The procedure’s requirement to introduce visual changes in the participant’s background ensures high accuracy in frame rate detection but adds complexity for the user during the video session. Practically, this requirement can be removed by leveraging the inherent dynamics within the video content if present. We are currently developing such an approach.

#### 4.4.2. Spatial Quality Assessment

In order to obtain ground truth spatial quality metrics, we capture lossless screen frames at a 1 s rate during the session. For each frame, we calculate quality scores using two no-reference image quality assessment models: BRISQUE [26] and PIQUE [27]. The BRISQE metric was acquired with the *brisqe()* function of the *image-quality 1.2.7* python 3.9 package. As there is no stock implementation of the PIQUE criterion in python, we used the MatLab 24.1 implementation of PIQUE to acquire this metric.

Both models provide quality scores ranging from 0 to 100, with lower values indicating fewer distortions. BRISQUE evaluates spatial quality based on natural scene statistics, while PIQUE assesses the perceptual aspects of local distortions in the spatial domain. For each model, the average score per time slot is used to create a quality score label. Table 6 shows the rating scale of the PIQUE and BRISQUE metrics used to assess the spatial quality rating for each time slot. The ratings are categorized as excellent, good, fair, poor, and bad. For PIQUE, the quality scale and respective score ranges are assigned based on the experimental analysis in [28]. For BRISQUE, due to the lack of a universally agreed-upon quality scale rating, we propose a linear quality scale.

### 4.5. Exploratory Data Analysis (EDA)

#### 4.5.1. Label Analysis

Next, we present an analysis of the collected dataset. Figure 3 illustrates the spatial quality characteristics through cumulative distribution functions (CDFs) for the BRISQUE score, PIQUE score, and BRISQUE- and PIQUE-based quality ratings, as presented in Table 6.

Figure 3a depicts the distribution of BRISQUE scores, showing two separate increases that indicate a distinction between two general quality groups obtained during the sessions. Correspondingly, the BRISQUE-based quality ratings are predominantly “average” (66%), aligning with the primary increase, while “poor” and “bad” ratings together account for 29%. The “good” quality rating is rare, comprising only 3%. The PIQUE distribution in Figure 3b shows a more continuous increase, indicating a more even distribution. For the PIQUE ratings, most samples are classified as “good” (52%) and “fair” (37%), while the remaining samples are rated as “poor” (11%). As the PIQUE-based quality rating is derived from experimental analysis, it offers a more reliable metric compared to the BRISQUE-based rating. Figure 4 presents the probability density distributions of BRISQUE, PIQUE, and frame rates under various bandwidth limitations, illustrating the strong correlation between spatial image quality and bandwidth in WhatsApp. Figure 4b shows that the PIQUE distribution for each bandwidth limit is concentrated around one or two central values. At 250 kBps, typically used by WhatsApp without network restrictions, the distribution is narrowly centered around a low value, indicating stable high quality. In contrast, at 125 kBps and 60 kBps, the distributions are centered around two distinct values for each bandwidth. This suggests different spatial adaptations employed by WhatsApp in response to bandwidth limitations. Figure 4c shows that the frame rate distribution has two primary peaks: The 250 kBps distribution is narrowly centered around 20 FPS; the 125 kBps distribution is more widely centered around 20 FPS; the 60 kBps, 30 kBps, and 15 kBps distributions are centered around approximately 15 FPS. This finding is consistent with our analysis of WhatsApp logs, which revealed that the intended frame rates for video sessions are primarily 15 or 20 FPS, with actual frame rates fluctuating around these target values. This analysis suggests that WhatsApp prioritizes maintaining stable frame rates of 15 to 20 FPS while adjusting spatial quality to accommodate varying bandwidth conditions. Under constant bandwidth conditions, WhatsApp efficiently maintains continuous video, with only a minor density below 10 FPS, indicating minimal occurrences of frame freezes.

Figure 5 presents the relationship between packet size and packet count with utilized bandwidth. Figure 5a demonstrates a logarithmic relationship between the packet size and utilized bandwidth. The majority of packets (82%) range between 900 and 1150 bytes, indicating consistent packet sizes for most of the utilized bandwidth range. Packet sizes below 700 bytes are infrequent (less than 5%) and are associated with low bandwidth utilization (below 40 kBps). Figure 5b demonstrates a linear relationship between the packet count and utilized bandwidth, exhibiting perfect correlations.

#### 4.5.2. Feature Correlation

From the correlation heat map, first, we performed a Pearson correlation [29] test of the extracted features, as presented in Figure 6. From this analysis, we see that there is a clear “blocking” of features, i.e., the *PIAT* features are inversely correlated to the *packet size* features. This phenomenon follows, as the size of the packet is in a direct proportion to the time it takes to transmit it over the line:
(6)
$$\rho_{X,Y}=\frac{Cov[X,Y]}{\sigma (X)\sigma (Y)}$$


The above equation measure ranges from 
[−1,1]
, where 1 indicates a perfect positive correlation, i.e., an increase in one feature is associated with an increase in the other, while 
−1
 represents a perfect negative correlation. Features with correlation values close to 
±1
 are the most informative, as they suggest that the feature (or its inverse) has a strong influence on the predicted variable.

#### 4.5.3. Feature Principle Component Analysis

Principal Component Analysis (PCA) [30] is a widely used technique for dimensionality reduction and the visualization of high-dimensional data. It works by identifying, in an iterative manner, the axes (called Principal Components or 
PC
s) along which the data exhibits the highest variance. These axes are derived through a linear combination of the original coordinate system.

The first Principle Component (
$PC_{1}$
) captures the direction of maximum variance in the data. Each subsequent 
PC
 is chosen to (1) account for the next higher variance, and (2) it should be orthogonal to all previously selected 
PC
s. The result is a new coordinate system where the components are uncorrelated, and the total variance is sorted in descending order across the components.

As may be concluded from Figure 7, a cumulative variance that exceeds 90% may be reached after the inclusion of the first 7 PCs, while a cumulative variance of greater than 99% is achieved only if we include the first 14 PCs. This analysis demonstrates that most of the features are necessary for the prediction of labels.

### 4.6. Feature Selection

To identify the most influential features, we employed Shapley analysis—a method derived from game theory—to quantify the contribution of each player 
i∈[1,N]
 (features in the dataset) in a cooperative game with respect to the final prize (the predicted label). The importance of each feature (i.e., its Shapley value) is calculated by averaging its marginal contribution over all possible cooperation possibilities (i.e., all possible ways a player *i* could join cooperation *S*) to the final model’s prediction.

For this matter, a predictive model *Y* is first trained, and then, for each sample in the dataset, the Shapley value is calculated with the following formula:
(7)
$$\phi_{i}=\sum_{S\subseteq \{1,\ldots N\}\setminus \{i\}}\frac{|S|!\cdot (N-|S|-1)!}{N!}\cdot \left[Y\left(S\cup \left\{i\right\}\right)-Y\left(S\right)\right]$$

where *N* is the total number of features, *S* represent the subset of features excluding feature *i* (i.e., the one in question), and 
|S|
 represent the size of subset *S*. 
Y(S∪{i})
 is the output of the model based on the features from *S*, including the feature we test here, while 
Y(S)
 represents the output of the model, excluding feature *i*. Intuitively, for each feature, we calculate the weighted gain this feature introduces into the final prediction of the model, and the final Shapley value is the average of this gain over all the possible ways this feature may be included in a cooperation instance with other features.

In our analysis, we highlight the importance of the number of packets as the most influential feature for all models and both settings (i.e., PIAT and packet size), as shown in Figure 8 and Figure 9. It should be noted that higher values (red color) are associated with larger increases in FPS and larger drops in BRISQEU and PIQUE, as for these QoE metrics, lower values are better, while for the FPS, an opposite dependence holds.

## 5. Experiments

In this section, we describe the procedures that used to test the proposed models. We divided the data by the feature type that was used for the training of the model into two types, viz., *PIAT* and *packet size* features.

For each of the five models, we used 10-fold Cross-Validation (CV), and we calculated the mean error of each predicted QoE with respect to the test set. The mean results over all 10 CV folds are summarized in Table 7. To test the importance of the two types of features that were extracted from the network pickup, we performed separate experiments for *PIAT* and *packet size*, and then, we performed experiments for *all* features taken together.

### 5.1. ML Models

In each CV run, the training data were standardized (i.e., applying 
$\frac{x-\mu }{\sigma }$
 on each feature), centering the data and cleaning outliers by removing all samples exceeding the mean by more than 
3·σ
. It is important to note that the test dataset was not cleaned of outliers, so the results in Table 7 describe a scenario of the realistic application of the method on unconstrained data.

### 5.2. LMM-Based Agent

In our experiments, we used the latest and strongest ChatGPT LMM version—ChatGPT 4o (Omni). We chose this model due to its ease of application and possible use in production via the provided python API. In this study, we wished to demonstrate an easy way to train a model using the ChatGPT web-based client.

#### Training Procedure

We trained the model in an interactive way, as listed below:Setting up the Context: As ChatGPT4o is a Large Multimodal Model (LMM), it may be applied for a multitude of tasks. To focus it on the task we are interested in, we perform what is called a “context setting” step Figure 10, in which we describe, in general words, the type of “expert” we need the agent to become in order to answer our queries in the best way.Training Query: As with other models, we need to provide the agent data to train the model. As presented in Figure 11, we provide the agent with two csv files—one with features only (PIAT, packet size, or all) and a file that contains the corresponding labels (i.e., BRISQE, PIQUE, FPS)—and explicitly ask it to carry out training on the provided dataset.Test Query: To test the predictions of the model used by the agent, we provide it with another csv file with features only and ask it to produce labels for each of the entries Figure 12, performing the same evaluation metric, as described in Equation (8), as for the other models.Evaluation: As for the other models, we performed 10-fold CV, calculating the MAEP Equation (8) for each of the 10 test sets and training the model again on the training data for each of the 10 folds.

## 6. Results

In this section, we present the results of the experiments conducted to evaluate the performance of the proposed models. All experiments were carried out using the dataset collected by our team, as described in Section 4.

To assess model performance, we computed the Mean Absolute Error Percentage (MAEP) between the predicted values and the ground truth labels, as defined in Equation (8).

Here, we present the results for the experiments that were conducted to test the proposed models. All experiments were performed using the data collected by our team, as described in Section 4. We evaluated the performance of the proposed methods with two no-reference QoE metrics, viz., BRISQUE and PIQUE, and one classical QoE evaluator, i.e, the FPS of the video, and we calculated the Mean Absolute Error Percentage (MAEP) of the predictions compared to the labels, as shown in Equation (8):
(8)
$$\epsilon^{MAEP}\left(t,p\right)=\left|100-\frac{p\cdot 100}{t}\right|$$

where *t* denotes the true label, and *p* denotes the predicted value.

We tested three feature settings via the following: (1) using only the features related to the packet inter-arrival times (PIATs); (2) using only features related to the packet sizes in traffic; and (3) using a combination of both. The results for all configurations and target metrics are summarized in Table 7.

## 7. Conclusions and Future Work

In the current work, we analyzed a large dataset collected in a controlled setting on the WhatsApp IMVCA. We trained four ML algorithms and proposed an alternative approach for easy and precise predictions by employing a ChatGPT4o model, which is an LMM-based agent, improving the MAEP of the RF model by 24.3%, 17.7%, and 25.7% according to the SOTA method used for QoE predictions under encrypted settings.

The main strength of the proposed method lies in its ease of application, as it requires neither high-speed computational equipment nor high power demands to produce a highly accurate model for QoE prediction.

Regarding the drawbacks of the method proposed, we observe its lack of interpretability. Due to the proprietary nature of ChatGPT4o, neither the model used by it to generate the predictions nor the accurate settings for existing models that can produce comparable results are disclosed, as they constitute the intellectual property (IP) of OpenAI, the company behind ChatGPT.

In addition, as the models are stored online, the inference time is longer compared to other methods described in Table 8, which makes this method less suitable for online applications.

As future work, we designate several lines of research as follows: (1) the exploration of other IMVCA applications under the encrypted traffic setting, as the applicability of our method to a specific application constitutes one of the main weaknesses of our work. (2) Another topic that may be further explored is the evaluation of other QoE metrics, such as the Mean Opinion Score (MOS). (3) Finally, other LMM agents besides ChatGPT may be employed in this domain.

## Figures and Tables

**Figure 1 sensors-25-04450-f001:**
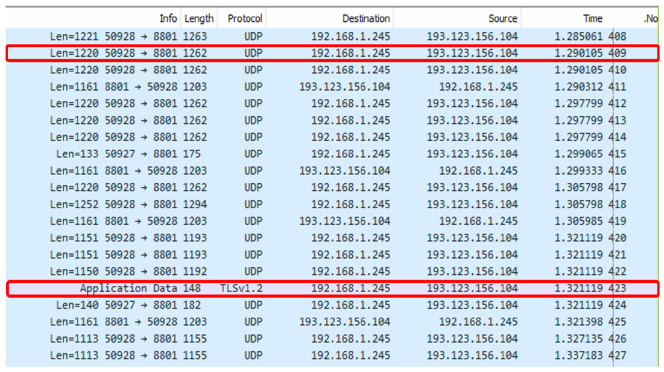
An example of feature extraction with the WireShark application in the lab environment. Red boxes highlight the traffic passed from one user encrypted by Transport Layer Security (TLS) protocol.

**Figure 2 sensors-25-04450-f002:**
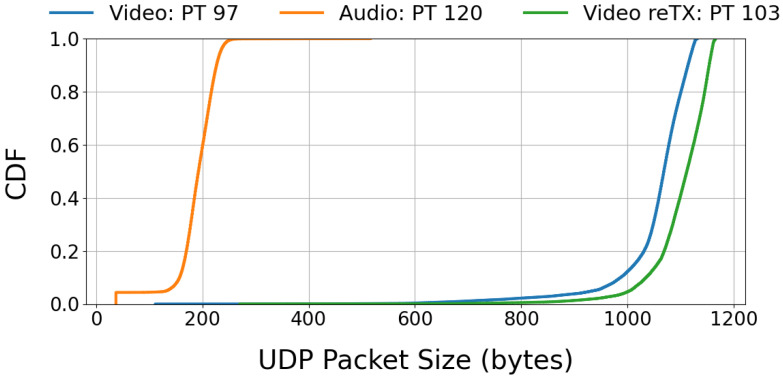
Cumulative distribution function (CDF) of UDP packet size by payload type for the WhatsApp dataset. PT denotes payload type, and reTX denotes retransmission.

**Figure 3 sensors-25-04450-f003:**
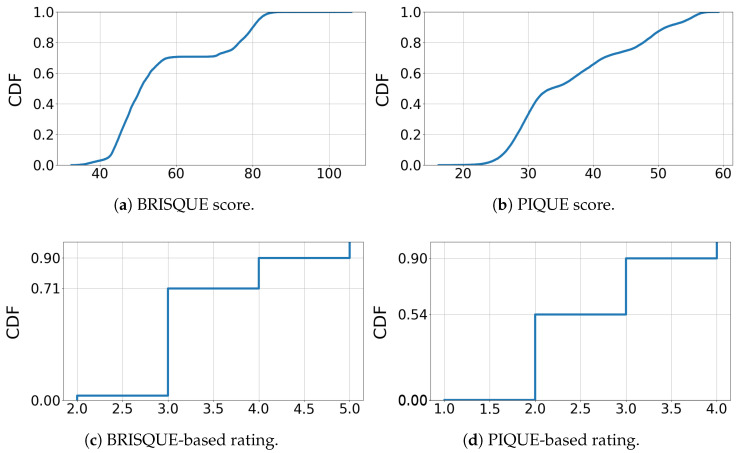
Characterization of the spatial quality through CDFs.

**Figure 4 sensors-25-04450-f004:**
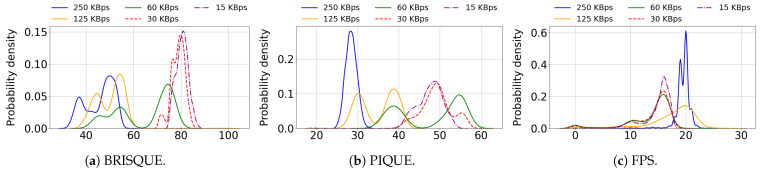
Density distribution of BRISQUE, PIQUE, and frame rate corresponding to each bandwidth limit: 250 kBps, 125 kBps, 60 kBps, 30 kBps, and 15 kBps.

**Figure 5 sensors-25-04450-f005:**
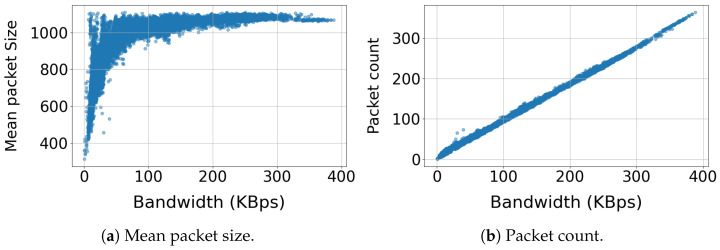
Relationship between bandwidth and (**a**) mean packet size and (**b**) packet count.

**Figure 6 sensors-25-04450-f006:**
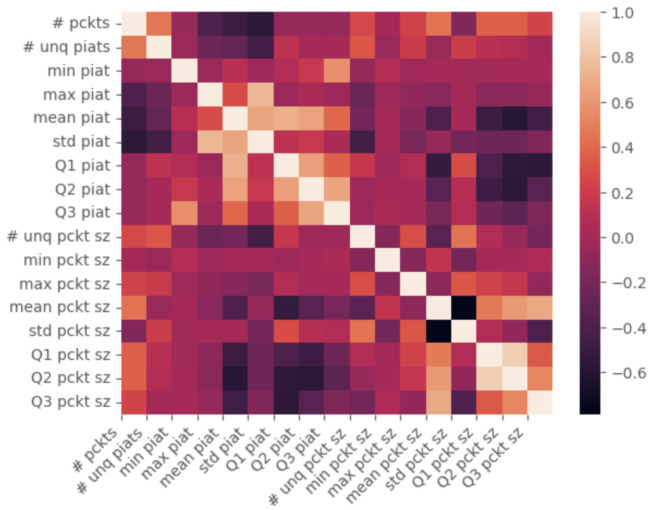
A heatmap representing the correlation of the features extracted from the PCAP files, as presented in Table 4.

**Figure 7 sensors-25-04450-f007:**
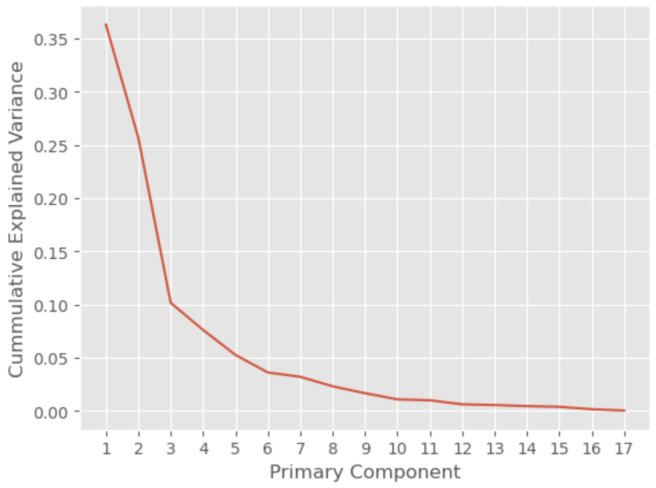
The PCA of the features in the dataset.

**Figure 8 sensors-25-04450-f008:**
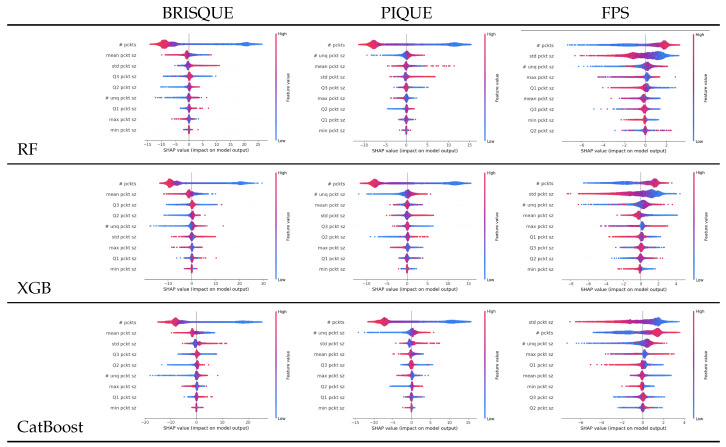
Shapley analysis for the three algorithms RF, XGB, and CatBoost on the packet size’s statistical features.

**Figure 9 sensors-25-04450-f009:**
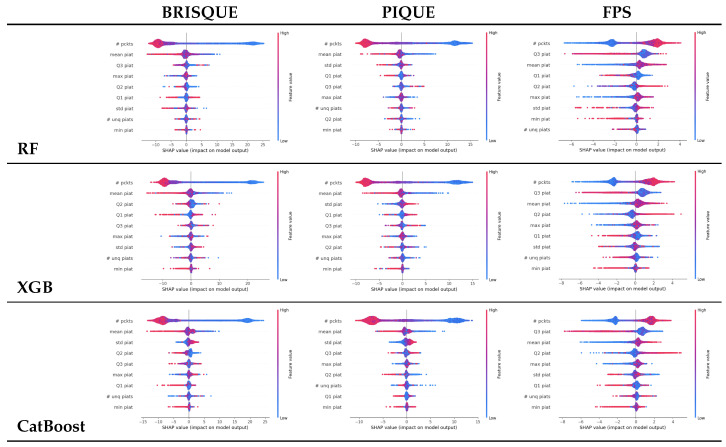
Shapley analysis for Random Forest, XGBoost and CatBoost on packet inter-arrival time (PIAT) features, as shown in Table 4.

**Figure 10 sensors-25-04450-f010:**
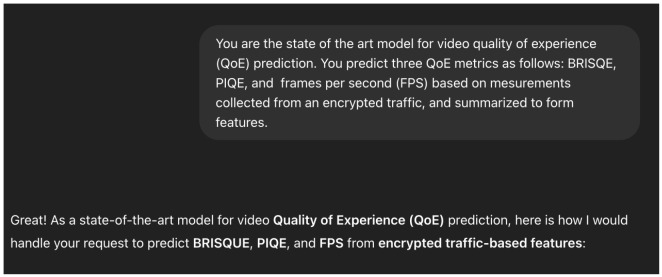
The initialization query of ChatGPT, which sets the context of the model.

**Figure 11 sensors-25-04450-f011:**
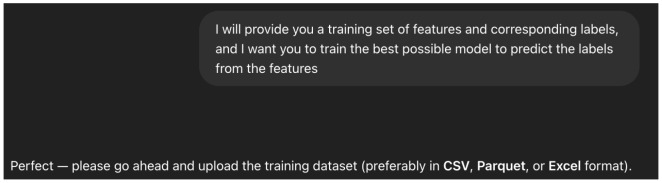
The query used for the training procedure for each of the CV folds.

**Figure 12 sensors-25-04450-f012:**
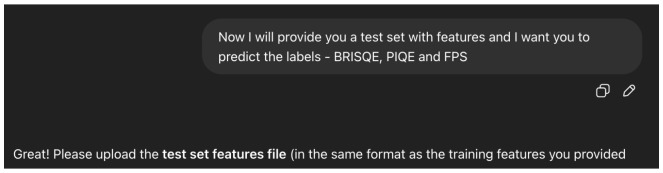
The query used for the test procedure for each of the CV folds.

**Table 1 sensors-25-04450-t001:** Description of the features that were recorded between the two participants in an IMVCA session.

Name	Abbreviation	Data Type	Description
frame.time_epoch	TE	float64	The arrival time of the current frame
frame.time_relative	RT	float64	The time passed since the start of the current time epoch
ip.src	IP_SRC	str	The source IP address of the packet
ip.dst	IP_DST	str	The destination IP address of the packet
ip.proto	IP_PROTO	float64	The protocol of the IP packet
ip.len	IP_LEN	float64	The length of the IP packet
udp.srcport	UDP_SRC_PRT	int64	The source port of the UDP datagram
udp.dstport	UDP_DST_PRT	int64	The destination port of the UDP datagram
udp.length	UDP_LEN	int64	The length of the UDP datagram

**Table 2 sensors-25-04450-t002:** Specifications of the sensors of the cameras used in the experiments.

Device	Resolution	Sensor Size (Inches)
Iphone SE (2020)	1.2 MP	1/3
Iphone 11	12 MP	1/2.55
Galaxy 9	8 MP	1/2.55
MacBook Air m1	720p	1/2.55
Asus ROG g17	720p	1/2.55
External Internet Camera	1080p	1/2.55

**Table 3 sensors-25-04450-t003:** The limitations imposed on the network line to simulate a realistic IMVCA session.

Limiting Factor	Description
Bandwidth limit	The maximal available bandwidth for the session was set to one of the following: 250, 125, 60, 30, or 15 kbps. The values were evenly distributed to avoidbias.
Bandwidth drops	A random drop in bandwidth from 250 to 15 kbps was introduced into the session.
Packet loss	The probability for losing a packet in the process of the session was chosen randomly and evenly distributed from 0%, 1%, 2%,5%, or 10%.

**Table 4 sensors-25-04450-t004:** The features extracted from the PCAP file of the IMVCA session. There are two types of features related to the size of the packets and to their inter-arrival timing.

Feature Type	Feature	Data Type	Description
General	#_pckts	int16	The number of packets in time window
Packet Size	#_unq_pck_sz	int16	Number of unique packet sizes in time window
	min_pckt_sz	int16	Minimal packet size in time window
	max_pckt_sz	int16	Maximal packet size in time window
	mean_pckt_sz	float32	Mean packet size in time window
	std_pckt_sz	float32	Standard deviation of the packet sizes in time window
	q1_pckt_sz	float32	Proportion of the packet sizes that are larger than 75% of the packet sizes
	q2_pckt_sz	float32	Proportion of the packet sizes that fall in range of 50–75% of the packet sizes
	q3_pckt_sz	float32	The lowest proportion of the packet sizes that are smaller than 25% of the packet sizes
PIAT	#_unq_piats	int16	Number of unique PIATs in time window
	min_piat	int16	Minimal PIAT in time window
	max_piat	int16	Maximal PIAT in time window
	mean_piat	float32	Mean PIAT in time window
	std_piat	float32	Standard deviation of the PIAT in time window
	q1_piat	float32	Proportion of the PIATs that are larger than 75% of the packet sizes in the data
	q2_piat	float32	Proportion of the PIATs that fall in range of 50–75% of the packet sizes in the data
	q3_piat	float32	The lowest proportion of the PIATs that are smaller than 25% of the packet sizes

**Table 5 sensors-25-04450-t005:** Classification Accuracy of WhatsApp Packets.

Actual	Classified		Packet Count	Average Packet Size (Bytes)
	**Non-Video**	**Video**		
Non-Video	99.93%	0.07%	285,751	187
Video	0.01%	99.99%	3,583,870	1054

**Table 6 sensors-25-04450-t006:** Quality scale ratings of PIQUE and BRISQUE.

PIQUE Range	BRISQUE Range	Rating
0–20	0–20	Excellent
21–35	21–40	Good
36–50	41–60	Fair
51–80	61–80	Poor
81–100	81–100	Bad

**Table 7 sensors-25-04450-t007:** Here, we present the performance of five different prediction models with respect to the task of QoE prediction expressed in MAEP Equation (8) on three popular QoE metrics, viz., BRISQUE, PIQUE, and FPS. The results are grouped by the type of features used for prediction, where we present the best results in each subgroup using bold scripts.

Features Type	Algorithm	BRISQUE	PIQUE	FPS
**Packet Size**	**Random Forest**	6.59±7.369	7.00±7.240	17.82±64.129
	**XGBoost**	7.51±7.981	8.10±8.029	19.62±71.339
	**CatBoost**	7.52±7.869	8.11±7.966	19.52±72.166
	**SVM**	7.51±7.981	8.10±8.029	22.42±96.570
	**LMM Agent**	**5.20** ± **0.067**	**5.67** ± **0.055**	**14.83** ± **1.222**
**PIAT**	**Random Forest**	7.73±8.060	8.06±7.845	25.22±71.662
	**XGBoost**	9.45±8.759	9.41±8.658	25.77±75.612
	**CatBoost**	9.05±8.280	8.88±7.922	24.92±76.855
	**SVM**	9.45±7.759	9.41±8.658	23.81±102.389
	**LMM Agent**	**4.61** ± **0.100**	**5.36** ± **0.119**	**13.24** ± **0.733**
**All**	**Random Forest**	**6.09** ± **6.543**	**6.52** ± **6.702**	19.55±61.043
	**XGBoost**	8.30±7.994	8.32±7.890	20.63±62.839
	**CatBoost**	7.52±7.169	7.69±7.209	18.87±62.004
	**SVM**	8.51±8.418	8.75±8.212	21.20±93.580
	**LMM Agent**	6.19±0.074	7.18±0.056	**16.51** ± **0.788**

**Table 8 sensors-25-04450-t008:** Inference time for different models and feature settings expressed in milliseconds.

Algorithm	Packet Size	PIAT	All
Random Forest	1.2±0.20	1.2±0.18	1.2±0.21
XGBoost	0.7±0.37	0.7±0.54	1.1±0.44
CatBoost	0.2±0.03	0.2±0.01	0.3±0.03
SVM	1.3±0.85	0.9±0.19	1.0±0.14
ChatGPT (4o) Agent	1.9±0.41	2.0±0.45	2.0±0.47

## Data Availability

The data and code are freely available on https://github.com/mchlsdrv/imvca_qoe_predictor (accessed on 13 July 2025).

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
