# Peer review of "Reliable QoE Prediction in IMVCAs Using an LMM-Based Agent"

_sensors, 2025, doi:10.3390/s25144450_

Round 1
Reviewer 1 Report
Comments and Suggestions for Authors
The authors investigate the possibility of predicting Quality of Experience (QoE) for WhatsApp video calls by using network traffic features, specifically PIAT and packet size. They trained and evaluated multiple machine learning models, including Random Forest and their own deep learning-based models (QoENet1D and QoEAE1D), as well as an approach based on a large language model (ChatGPT-4o). The models were tested on a dataset of 720 samples using 10-fold cross-validation, and prediction quality was assessed using BRISQUE, PIQE, and FPS metrics.
My comments on the paper relate to the following points:
- The explanation of the formula 2:
I suggest that the authors explain the meaning of the term dk in the attention equation since the other components (Q, K, V) have already been defined. This would help the reader better understand the purpose of scaling the dot product by dk, which is important for the stability of the softmax function
- The conclusion section, which I believe should be expanded for a more thorough discussion.
The conclusion section summarizes the key findings, but it should be expanded to provide a more comprehensive discussion of the approach’s limitations, the models’ contributions compared to existing methods, and the potential applicability beyond the WhatsApp environment. Furthermore, a deeper reflection on the implications of these results for the future development of non-intrusive QoE assessment methods in encrypted traffic scenarios would strengthen the paper.
Author Response
Dear Reviewer,
Thank you for your insightful and constructive remarks. We have carefully considered your feedback and made the necessary revisions in the new version of our manuscript.
To further clarify the changes, we have also included a supplementary PDF file that provides a detailed explanation of the modifications implemented in response to your comments.
We sincerely appreciate your time and effort in reviewing our work.
Warm regards,
The Team

Reviewer 2 Report
Comments and Suggestions for Authors
- Line 201 of the article describes the data collection process but lacks detailed explanation of data annotation. For example, how were BRISQUE and PIQE annotated? Were there any subjective factors involved? It is recommended to provide more detailed information to improve the credibility and reproducibility of the results.
- In line 179 of the article, it mentions the use of ChatGPT as the LMM model, but does not explain the reasons for choosing ChatGPT or whether other LMM models have been tried. Additionally, there is a lack of detailed description of the LMM model training process, such as training time and parameter settings. It is recommended to supplement relevant information to enhance the interpretability of the results.
- The article utilizes multiple features but does not perform feature selection and importance analysis. It is recommended to use feature selection methods, such as L1 regularization or feature ranking, to identify the most important features for QoE prediction and analyze their mechanisms of action.
- Although there are related work reviews (such as YouTube QoE prediction), the experimental part does not quantitatively compare with these existing methods, nor does it use benchmark datasets (such as the LIVE video database) to enhance the generality and comparability of the research. The authors are suggested to supplement comparative experiments or provide reasons why this is not possible.
- The article mentions using a dataset containing only 720 samples and performing 10-fold cross-validation. Although the results show that the LMM agent performs well, the variance of the results is extremely small (e.g., PIQE error is only ±0.055), which differs significantly from other models, possibly indicating issues with the evaluation process or unexplained sources of randomness. Suggestions: 1) Report more statistical indicators (such as confidence intervals); 2) Further verify the statistical significance of the results (e.g., using t-tests); 3) Expand the dataset.
Author Response

(The authors gave the same response as above.)

Reviewer 3 Report
Comments and Suggestions for Authors
This study addresses the challenge of QoE prediction for IMVCAs from encrypted traffic, innovatively applying Large Multimodal Models (LMMs) to infer user experience for ISPs. The experimental design is comprehensive, and the open-source dataset (on GitHub) adds significant academic value. The LMM-based agent outperforms traditional ML algorithms (RF/XGBoost) with 17.7%-25.7% lower errors in BRISQUE/PIQE/FPS metrics, demonstrating its effectiveness. However, the research requires improvements in theoretical depth, scenario generalizability, and model interpretability. Detailed comments are provided below.
-For the part of methodology, the proposed method should be described in detail. The existing methods such as random forest, SVM should be deleted.
-The paper lacks clarification on how LMMs capture the logic between temporal network features (e.g., how attention mechanisms associate "packet size fluctuations with video"). SHAP value analysis or attention distribution visualization is recommended to enhance interpretability.
-The paper does not compare with recent cross-modal models (e.g., Flamingo, Kosmo-2) or specialized temporal LMMs (e.g., Time-LLM), making it hard to confirm the optimality of ChatGPT-4o.
-Real-time performance (e.g., single-sample inference latency) is unvalidated, which is critical for industrial QoE monitoring.
Author Response

(The authors gave the same response as above.)
